# Prolonged Carriage of Carbapenemase-Producing *Enterobacteriaceae*: Clinical Risk Factors and the Influence of Carbapenemase and Organism Types

**DOI:** 10.3390/jcm10020310

**Published:** 2021-01-15

**Authors:** Yong Kyun Kim, In Bok Chang, Han Sung Kim, Wonkeun Song, Seung Soon Lee

**Affiliations:** 1Division of Infectious Diseases, Department of Internal Medicine, Hallym University Sacred Heart Hospital, Hallym University College of Medicine, Anyang 14068, Korea; amoureuxyk@hallym.or.kr; 2Department of Neurosurgery, Hallym University Sacred Heart Hospital, Hallym University College of Medicine, Anyang 14068, Korea; 3Department of Laboratory Medicine, Hallym University Sacred Heart Hospital, Hallym University College of Medicine, Anyang 14068, Korea; kimhs@hallym.or.kr; 4Department of Laboratory Medicine, Hallym University Kangnam Sacred Heart Hospital, Hallym University College of Medicine, Seoul 07441, Korea; swonkeun@hallym.or.kr; 5Division of Infectious Diseases, Department of Internal Medicine, Hallym University Chuncheon Sacred Heart Hospital, Hallym University College of Medicine, Chuncheon 24253, Korea

**Keywords:** carbapenemase-producing *Enterobacteriaceae*, carbapenemase, organism, prolonged carriage

## Abstract

Prolonged carriage of carbapenemase-producing *Enterobacteriaceae* (CPE) constitutes a substantial epidemiologic threat. This study aimed to evaluate whether the types of carbapenemase and organism can affect the duration of carriage and to evaluate the clinical factors associated with prolonged carriage. We retrospectively reviewed data for patients admitted between May 2013 and August 2018 who were identified as CPE carriers. A total of 702 patients were identified; the major types of carbapenemase and organism were Oxacillinase (OXA)-48-like (*n* = 480, 68.4%) and *Klebsiella pneumoniae* (*K. pneumoniae*) (*n* = 584, 83.2%). The analyses of time to spontaneous decolonization using the Kaplan–Meier method showed that OXA-48-like and *K. pneumoniae* were significantly associated with prolonged carriage (log rank, *p* = 0.001 and *p* < 0.001). In multivariable logistic analysis to assess the risk factors for CPE prolonged carriage in the 188 patients with available follow-up culture data for 3 months, *K. pneumoniae* (adjusted odds ratio [aOR] 6.58; 95% confidence interval [CI], 1.05–41.27; *p* = 0.044), CPE positive clinical specimen (aOR 11.14; 95% CI, 4.73–26.25; *p* < 0.001), and concurrent *Clostridioides difficile* infection (CDI) (aOR 3.98, 95% CI 1.29–12.26; *p* = 0.016) were predictive of prolonged carriage. Our results suggest that CP-*K. pneumoniae* may have higher probability of prolonged carriage, while the effect of OXA-48-like CPE is inconclusive. Furthermore, patients with CP-*K. pneumoniae* who had positive clinical specimen or concurrent CDI can cause a vicious circle in prolonged carriage.

## 1. Introduction

The occurrence and spread of carbapenemase-producing *Enterobacteriaceae* (CPE) have been increasing worldwide [1]. Outbreaks of CPE attributed to plasmid-borne genetic elements have been reported mainly in the healthcare setting [2]. However, transmission of CPE even in the community has already been demonstrated [3]. Colonization or carriage of carbapenem-resistant *Enterobacteriaceae* (CRE) may be followed by subsequent infections, which results in increased risk of death, especially in vulnerable patient groups [4,5]. Compared with non-CPE, CPE may play a more active role in outbreaks as an important reservoir for transmission [6,7]. Moreover, the carriage state of CPE can lead to higher risks of subsequent infections associated with worse outcomes than that of non-CPE [8,9].

In this regard, prolonged carriage of CPE both in the hospital setting and community constitutes a substantial epidemiologic threat. Active surveillance of CPE for early detection with the subsequent isolation of patients is an important component of the infection control bundle [10]. Furthermore, individualized active surveillance based on a risk-based approach has been recommended and considered as an effective infection control measure [11]. However, there are heterogenous screening practices in each institutions and high costs as well as limited isolation capacity are potential barrier to optimal preventive measures for CPE transmission [12,13,14,15]. 

Therefore, targeted screening algorithm based on predictive factors of prolonged CPE carriage may help to prioritize the limited resources. In addition, the risk stratification based on patients at high risk for prolonged carriage of CPE may help targeted implementation of infection control measures and development of evidence-based decolonization strategies. While there have been few reports comparing the clinical outcomes and transmission rates between patients colonized with different CPE [16,17], there are few clinical data related to risk factors associated with prolonged CPE carriage and comparative studies of the influence of carbapenemase type and underlying organism on CPE colonization duration. In the present study, we aimed to evaluate whether the types of carbapenemase and organism can affect the duration of carriage, and the clinical factors associated with prolonged carriage of CPE.

## 2. Materials and Methods

### 2.1. Study Design and Population

This retrospective study was conducted at Hallym University Sacred Heart Hospital, an 829-bed university-affiliated hospital in Anyang, Korea. We reviewed medical charts of patients admitted between May 2013 and August 2018 who were identified as CPE carriers via either rectal swabs or clinical cultures. To evaluate the clinical factors associated with prolonged carriage of CPE, we analyzed the data of patients with the opportunity for at least 3 follow-up cultures and available follow-up culture data for 3 months after the initial detection of CPE.

Initial detection was defined as the earliest date of confirmed evidence of CPE in an index culture study. During the study period, universal active surveillance cultures of CRE via rectal swab were performed weekly on hospitalized patients who were admitted to intensive care units (ICUs) because of constant challenges from admission of patients colonized with CRE in ICUs of our hospital. The weekly follow-up screening after the initial detection was continued during entire hospitalization including non-ICU settings until three consecutive negative follow-up cultures were collected and analyzed. In addition, weekly surveillance culture was performed on patients in general wards when CPE was identified via either rectal swabs or clinical cultures. When CPE was identified on clinical cultures, surveillance cultures of CPE on both clinical cultures and rectal swab cultures were followed up weekly during the entire hospitalization until three consecutive negative follow-up cultures were collected and analyzed. After discharge from hospital to the home or another hospital, they were recommended to be followed at least on a monthly basis at the outpatient clinic, and they underwent follow-up CRE cultures in a standardized manner.

### 2.2. Definitions and Clinical Data

Detailed definitions and data collection have been presented in our previous work [18]. Prolonged carriage and persistent carriage of CPE were defined as identification of CPE in either rectal swab or clinical culture for more than 3 and 6 months after initial detection, respectively. 

A clinical positive culture was defined as every clinical specimen positive for CPE other than rectal swab, regardless of clinical signs of infection. Spontaneous decolonization of CPE was defined as three consecutive negative weekly surveillance cultures that were not followed by any positive cultures. When CPE was identified on clinical cultures, three consecutive negative weekly follow-up clinical cultures, as well as rectal swab cultures, were required to document CPE clearance. The time to spontaneous decolonization was defined as the time interval from initial detection of CPE until three consecutive negative cultures were obtained. The Charlson comorbidity index score was used to evaluate comorbidity. Immunosuppressive therapy was defined as use of corticosteroid (≥20 mg of prednisolone or equivalent over 3 weeks, or ≥10 mg/day for at least 3 months), chemotherapy or radiotherapy, or other recognized T cell immunosuppressants such as tumor necrosis factor-alpha blockers and calcineurin inhibitors during the 3 months of follow-up. Data regarding the presence of any invasive catheter (central venous catheter, urinary catheter, enteral feeding tube, and tracheostomy) and systemic antibiotic use (administration for more than at least 48 h after CPE detection) during the 3-month follow-up period were collected. ICU admission (residence for at least 48 h), readmission (defined as hospitalization after discharge from the index hospitalization), and duration of hospitalization before readmission during the 3-month follow-up period were recorded. We also reviewed cocolonization of patients with CPE and vancomycin-resistant *Enterococci* (VRE) as well as concurrent *Clostridioides difficile* (*C*. *difficile*) infection (CDI), defined as the presence of symptoms such as diarrhea and either a positive stool test result for *C. difficile* toxins or colonoscopic findings demonstrating pseudomembranous colitis, during the 3-month follow-up period.

### 2.3. Microbiologic Methods

Specimens including rectal swabs were collected from patients during the study period. Rectal swabs were plated on primary CHROMagar KPC medium (CHROMagar, Paris, France) for direct screening of CRE. Other clinical specimens except rectal swabs were plated on blood agar (Becton-Dickinson, Sparks, MD, USA) and MacConkey agar (Becton-Dickinson). The agar plates were incubated at 36.5 °C in ambient air for 16–24 h. Bacterial identification was done with VITEK^®^ 2 system (bioMerieux, Marcy I’Etoile, France). Susceptibility testing to antibiotics including carbapenem was performed using the VITEK^®^ 2 system according to the Clinical and Laboratory Standards Institute M100S guidelines [19]. Isolates were tested further to detect carbapenemase production using the modified Hodge test and carbapenemase inhibition test [20]. In addition, we performed a polymerase chain reaction assay targeting the genes encoding *bla*_KPC_, *bla*_NDM_, *bla*_IMP,_
*bla*_VIM_, *bla*_GES_, and *bla*_OXA-48_ to confirm carbapenemase [21].

### 2.4. Statistical Analysis

The time to spontaneous decolonization within 90 days was assessed using the Kaplan–Meier method stratified by carbapenemase and organism types, and the probabilities of prolonged carriage during follow-up period were compared using the log-rank test.

In the selected patients with available follow-up culture data for 3 months after the initial detection of CPE, we compared various characteristics of patients with prolonged carriage with those of patients who experienced spontaneous decolonization within 3 months. In addition, we performed analyses among the patients with the most prevalent type of organism and carbapenemase in this study to evaluate the risk factors for prolonged carriage. Categorical variables were compared using the *χ*^2^ or Fisher’s exact test, and continuous variables were compared using the Mann–Whitney *U* test, as appropriate. To evaluate risk factors for prolonged carriage of CPE, all variables with *p* value < 0.1 in the univariable analysis were included in a multivariable logistic regression model. All statistical analyses were performed using SPSS for Windows software package, version 21 (SPSS Inc, Chicago, IL, USA). All tests of significance were two-tailed and a *p*-value < 0.05 was considered to indicate statistical significance.

## 3. Results

### 3.1. Study Patients and Identified Isolates

We identified 702 CPE colonized patients during the study period. The majority was Oxacillinase (OXA)-48-like CPE (*n* = 480, 68.4%), followed by New-Delhi-Metallo-beta-lactamase (NDM) CPE (*n* = 126, 17.9%) and *Klebsiella pneumoniae* carbapenemase (KPC) CPE (*n* = 96, 13.7%). 

There were 659 patients with monobacterial colonization and 43 patients with polymicrobial carriage (more than one isolated species of CPE had the same carbapenemase gene in the same patient). Among patients with monobacterial carriage, the result of species identification revealed that *Klebsiella pneumoniae* (*K. pneumoniae*) was the most prevalent CPE isolates (*n* = 545, 82.7%). The other organisms include 50 isolates (7.6%) of *Enterobacter* spp., 20 isolates (3.0%) of *Serratia marcescens* (*S. marcescens*), 17 isolates (2.6%) of *Escherichia coli* (*E. coli*), 16 isolates (2.4%) of *Citrobacter freundii* (*C. freundii*), and 11 isolates (1.7%) of *Klebsiella oxytoca* (*K. oxytoca*). There were 43 patients with polymicrobial carriage, 39 of whom had *K. pneumoniae* with other organisms (15 with *E. coli*, 12 with *C. freundii*, 9 with *Enterobacter* spp., 1 with *S. marcescens*, 1 with *E. coli*, *C. freundii*, and *Enterobacter* spp., 1 with *E. coli* and *Enterobacter* spp.) and the other four had *Enterobacter* spp. with *C. freundii*.

### 3.2. Probability of Prolonged Carriage by Different Types of Carbapenemase and Organism

The study patients were divided into 3 subgroups based on the carbapenemase types (OXA-48-like, KPC, NDM), and 2 subgroups based on the organism types (*K. pneumoniae* and others). Kaplan–Meier analysis showed that patients colonized with OXA-48-like CPE were significantly associated with prolonged carriage than those colonized with other types of carbapenemase (log rank, *p* = 0.001) (Figure 1A). When stratifying the risk of prolonged carriage by carbapenemase types, OXA-48-like CPE showed the significantly increased risk of prolonged carriage than NDM CPE (*p* < 0.001), and there was no significant difference between OXA-48-like CPE and KPC CPE (*p* = 0.235) (Figure 1B). In addition, KPC CPE showed a tendency toward an increased rate of prolonged carriage than NDM CPE (*p* = 0.056) in our analysis.

Figure 2A shows the Kaplan–Meier curves for probability of prolonged carriage in patients who were and were not colonized with carbapenemase-producing *K. pneumoniae*. *K. pneumoniae* group was significantly associated with prolonged carriage compared with non-*K. pneumoniae* group (*p* < 0.001). When the analysis was performed in patients only with monobacterial carriage, carbapenemase-producing *K. pneumoniae* group was significantly associated with prolonged carriage (*p* = 0.001) (Figure 2B).

### 3.3. Risk Factor Analysis for Prolonged Carriage of CPE

Of the study patients, 514 did not have available follow-up culture data after 3 months of initial detection (194 died and 320 were lost to follow-up within 3 months after initial detection). Finally, we included the 188 selected patients that had available follow-up data for 3 months after initial detection of CPE to evaluate the risk factors associated with prolonged carriage of CPE (Figure 3). Of these patients, 83 (44%, 83/188) had prolonged carriage after 3 months of initial detection. The rates of prolonged carriage among different types of CPE were highest in OXA-48-like CPE (51%, 59/116), followed by KPC CPE (46%, 15/33) and NDM CPE (23%, 9/39). Among 31 patients with available follow-up culture data after 6 months of initial detection, 20 (11%) remained culture positive for CPE after 6 months (persistent carriage).

The identification of CPE was found in 106 patients of rectal swab culture only and 82 patients of both rectal swab culture and clinical culture (46 sputum culture, 41 urine culture, 14 surgical wound culture, 9 blood culture, 7 intra-abdominal fluid culture). There were 27 patients with clinical positive culture, including bacteremia, treated for suspicious CPE infections. We found that 11 of 27 patients had initial detection of CPE in clinical specimens. All of rectal swab cultures performed on the day of detection of clinical positive cultures revealed positive for CPE, which indicated the initial colonization prior to clinical infections. The result of species identification revealed that there were 164 patients with *K. pneumoniae*, 26 patients with *Enterobacter* spp., 17 patients with *C. freundii*, and 14 patients with *E. coli*, including 30 patients with polymicrobial carriage. The proportion of carbapenemase types carried by 164 isolates of *K. pneumoniae* were OXA-48-like gene (69.5%, 114/164), NDM gene (10.4%, 17/164), and KPC gene (20.1%, 33/164), which indicated that 98.3% (114/116) of OXA-48-like gene and 100% (33/33) of KPC gene were carried by *K. pneumoniae*, respectively. All of 14 isolates of *E. coli* carried OXA-48-like gene (100%, 14/14), whereas most of *Enterobacter* spp. and *C. freundii* carried NDM gene (80.8%, 21/26 of *Enterobacter* spp. and 70.6%, 12/17 of *C. freundii*).

Table 1 shows the differences in demographics and clinical characteristics between the 105 patients who experienced spontaneous decolonization within 3 months and the 83 patients who remained culture positive after 3 months. There were several statistically significant differences in clinical factors and comorbidities between the two groups. Univariable analysis revealed that the OXA-48-like type was associated with prolonged carriage (*p* = 0.019). In contrast, the NDM type had the opposite result that indicated shorter duration of carriage (*p* = 0.003). The influence of the underlying bacterium carrying carbapenemase gene revealed that polymicrobial carriage (*p* = 0.007), *K. pneumoniae* (*p* < 0.001), and *E. coli* (*p* < 0.001) were significantly associated with prolonged carriage of carbapenemase gene. In contrast, *Enterobacter* spp. had the opposite result that indicated shorter duration of carriage (*p* = 0.006). VRE co-colonization (*p* = 0.007), concurrent CDI (*p* = 0.002), longer duration of hospitalization (*p* < 0.001), and clinical positive culture (*p* < 0.001) were significantly associated with prolonged carriage of CPE after 3 months of initial detection. In addition, exposure to some particular systemic antibiotics, including fluoroquinolones (*p* = 0.015), carbapenem (*p* < 0.001), tigecycline (*p* = 0.003), and glycopeptide (*p* = 0.003) were found to be significantly associated with prolonged carriage of CPE.

In the multivariable analysis, clinical positive culture (adjusted odds ratio [aOR], 11.14; 95% confidence interval [CI], 4.73–26.25; *p* < 0.001), concurrent CDI (aOR, 3.98; 95% CI, 1.29–12.26; *p* = 0.016), and longer duration of hospitalization (aOR, 1.01; 95% CI, 1.004–1.03; *p* = 0.008) were suggestive of significant clinical risk factors for prolonged carriage of CPE (Table 2). In addition, *K. pneumoniae* (aOR, 6.58; 95% CI, 1.05–41.27; *p* = 0.044) and *E. coli* (aOR, 13.48; 95% CI, 1.18–153.48; *p* = 0.036) that carried carbapenemase gene were significantly associated with an increased risk of prolonged carriage. Exposure to carbapenem (OR, 2.32; 95% CI, 0.97–5.55; *p* = 0.06) showed a tendency toward an increased risk of prolonged carriage of CPE, despite the lack of statistical significance.

Table 3 and Table 4 described the clinical factors predictive of prolonged carriage among the selected patients with OXA-48-like-producing CRE and carbapenemase-producing *K. pneumoniae* (the most prevalent type of carbapenemase and organism in this study). Among patients with OXA-48-like-producing CRE, multivariable analysis revealed that clinical positive culture (aOR, 13.93; 95% CI, 4.53–42.81; *p* < 0.001), concurrent CDI (aOR, 10.49; 95% CI, 2.01–54.88; *p* = 0.005), and exposure to carbapenem (aOR, 8.96; 95% CI, 2.24–35.83; *p* = 0.002) were suggestive of significant risk factors for prolonged carriage. Among patients with carbapenemase-producing *K. pneumoniae*, multivariable analysis revealed that clinical positive culture (aOR, 10.40; 95% CI, 4.45–23.34; *p* < 0.001), concurrent CDI (aOR, 4.38; 95% CI, 1.37–13.94; *p* = 0.013), exposure to carbapenem (OR, 2.95; 95% CI, 1.23–7.06; *p* = 0.015), and longer duration of hospitalization (aOR, 1.01; 95% CI, 1.00–1.03; *p* = 0.009) were suggestive of significant risk factors for prolonged carriage.

## 4. Discussion

In the present study, we evaluated whether the types of carbapenemase and organism can affect the duration of carriage, and the clinical factors associated with prolonged carriage of CPE. Interestingly, the duration of CPE colonization was affected by types of carbapenemase and organism. The probability of prolonged carriage estimated by the Kaplan-Meier method showed that OXA-48-like-producing CPE and *K. pneumoniae* were more likely to have prolonged carriage compared to NDM CPE and other species such as *Enterobacter* spp., respectively. In addition, multivariable logistic analysis to assess the strongest independent risk factor for prolonged carriage of CPE revealed that *K. pneumoniae* was significantly associated with prolonged carriage, although OXA-48-like CPE was not significantly associated with prolonged carriage. Furthermore, the results indicate that clinical positive culture, concurrent CDI, and longer duration of hospitalization were predictive of prolonged carriage of CPE. 

Data regarding the natural duration of CPE are scarce [22]. Information on natural duration of carriage and risk factors for prolonged carriage of CPE may help institutions organize preventive strategies including preemptive contact precautions and active surveillance culture for high-risk patients with CPE colonization in resource-limited settings. Previous data regarding the duration of CPE carriage described mainly KPC CPE cases [18,23,24,25,26]. However, little is known about the duration of carriage of OXA-48-like CPE and NDM CPE [27,28,29]. To the best of our knowledge, this is the first study to evaluate the clinical risk factors of prolonged carriage of OXA-48-like CPE with the largest number of cases (*n* = 116). Our results indicated that there were similar trends of prolonged carriage of OXA-48-like CPE and KPC CPE. In addition, it was interesting that NDM CPE had lower likelihood of prolonged carriage than OXA-48-like and KPC CPE, which was consistent with the results of one previous comparative study of NDM and KPC CPE [27]. The results of this study have implications in that the risk of prolonged carriage may differ between carbapenemase types and would be of great help in developing evidence-based CPE prevention strategies by predicting the duration of CPE carriage.

The influence of the underlying bacterium carrying the carbapenemase gene may be also important, and analysis assessing the interaction between carbapenemase type and organism type should be considered. Previous studies to evaluate the duration of CPE included mainly patients with carbapenemase-producing *K. pneumoniae* [18,23,24,25,26], and the most prevalent CPE isolate was *K. pneumoniae* (95.1%) in KPC group, while only 35.8% of NDM group had *K. pneumoniae* in one comparative study regarding the clearance of NDM and KPC CPE [27]. Therefore, we performed analyses the clinical risk factors associated with prolonged carriage in the selected patients with carbapenemase-producing *K. pneumoniae*, the most prevalent CPE isolates, and the risk factors were consistent with those in the entire study subjects. Meanwhile, our results of the multivariable analysis of risk factors for prolonged carriage revealed that *E. coli* was also significantly associated with prolonged carriage. In addition, polymicrobial carriage was not significantly associated with prolonged carriage in multivariable analysis, despite the statistical significance in univariable analysis. However, the clinical implication is uncertain, because of the small number of patients colonized with *E. coli* and different species producing the same carbapenemase gene at the same time. We believe that an additional multicenter study with large scale is fundamental to provide the precise information about potential impact of them.

An important finding of the present study was that patients with any clinical specimen positive for CPE had increased risk of prolonged carriage of CPE, which may be explained by a larger inoculum or burden of CPE. Regardless of the type of carbapenemase, clinical positive culture was consistently a significant risk factor for prolonged carriage of CPE in our study. This result may highlight the importance of active surveillance culture as the key component of infection control measures of CPE to identify asymptomatic colonizers earlier and prevent the development of subsequent clinical infection. Current strategies to apply active surveillance culture and contact precaution may differ in the context of CPE prevalence and endemicity [30]. However, there is a consensus that each institution should define high risk patients to identify CPE at admission using their own human or material resources [11]. We posit that our result implies that patients who previously tested positive for CPE in clinical specimens should be included as high-risk patients for CPE screening. Moreover, preemptive isolation and empiric contact precaution should be considered preferentially in patients with clinical positive culture results for CPE, even in resource-limited settings [31]. The present article has important clinical implications for future research such as case-control study for risk factors of clinical positive culture for CPE in patients with fecal carriage of CPE, and clinical trial to evaluate whether FMT can be performed preferentially among patients carrying CPE in clinical cultures as well as rectal swab cultures. 

Notably, our result indicated that concurrent CDI, suggestive of gut microbiota disruption following the administration of antibiotics, had a significant association with prolonged carriage of CPE. This result suggests that the disruption and alteration of intestinal microbiota by exposure to antibiotics may affect the activity of CPE, as several previous studies have revealed [32,33]. A previous study revealed that patients with recurrent CDI had a greater number and diversity of antibiotic resistance genes exclusively among Proteobacteria and in a small number of Firmicutes [34]. More recently, one study indicated that the characteristics of gut microbiota in patients with KPC CPE colonization were different from those in patients without CPE colonization [35]. Beyond its role as a treatment option for recurrent CDI, we believe that FMT for restoring gut microbial homeostasis may play a role as an infection prevention measure for CPE in the context of a decolonization strategy [36,37]. Furthermore, future research may consider performing FMT as a priority among CPE carriers with concurrent CDI at high risk for prolonged carriage of CPE. 

We also found that carbapenem use was associated with prolonged carriage of CPE, which was consistent with and can be reinforced by the previous study [22]. Although there is a paucity of data for evaluating the impact of individual antibiotics on the intestinal microbiota [38,39], a recent publication showed that carbapenem use was independently predictive of high relative abundance of KPC-producing *K. pneumoniae* in the gut microbiota [40]. Interestingly, one epidemiologic study revealed that the incidence of CRE may rise with increased hospital-wide carbapenem use over time [41]. From our point of view, a multifaceted infection control program for CPE should include targeted carbapenem stewardship that has already been used in the control of CPE outbreaks [2]. In addition, CPE colonized patients with use of carbapenem, presumably at high risk for prolonged carriage of CPE, may be a priority group for performing FMT.

This study has several limitations. First, this was a single center study; therefore, there can be other factors that can cause variability in results in other institutions. In addition, the geographic variability in distribution of CPE genes may decrease the external validity of this information. However, it was noteworthy that the clinical variables predicting prolonged carriage of CPE in our study were similar with those in other previous studies that evaluated patients with KPC CPE, although we included OXA-48-like CPE with the largest number of cases. Second, the retrospective nature of our present analyses may have resulted in information bias or missing data that influenced the results. We need to address that some patients did not have equal opportunities for assessment of decolonization in terms of number of total cultures and frequency of cultures (weekly during hospitalization, but not after discharge from hospital), which could introduce ascertainment bias because of the deviation from a standardized assessment. In addition, a considerable proportion of patients were lost to follow-up or died, which might have confounded the results of the risk factor analysis and made the research design (the Kaplan–Meier method) not ideal for the assessment of prolonged carriage. Further well-designed prospective study in large scale to assess the impact of types of carbapenemase and organism on prolonged carriage of CPE by use of the multivariable Cox proportional hazard regression model after adjustment of clinical variables is clearly warranted. However, we gathered extensive clinical information and the infection control protocols were consistent during the study period. Third, we only used rectal swab cultures for surveillance which might have insufficient sensitivity. The use of CPE real-time polymerase chain reaction assay with increased sensitivity to define spontaneous decolonization could have influence on the results of the duration of colonization and risk factors for prolonged carriage of CPE. Fourth, molecular determination of the exact type of carbapenemase gene using MLST and entire genome sequencing of each strain was not performed in our study, although the genetic support may have different significations. We believe that future research to evaluate the clonal spread of CPE using a technique in molecular biology may help regional coordination of CPE surveillance and control measures. In addition, we believe that it would be worthwhile to include gut microbiome variables in the future in predicting prolonged carriage of CPE [35,40]. Despite these limitations, our study has some strength in that we included a relatively large number of patients to evaluate the clinical variables affecting prolonged carriage of distinct types of CPE.

## 5. Conclusions

Our results suggest that CP-*K. pneumoniae* may have higher probability of prolonged carriage, while the effect of OXA-48-like CPE on prolonged carriage is inconclusive. In addition, patients with CPE in clinical specimens who are using carbapenem or have concurrent CDI can cause a vicious circle in prolonged carriage of CPE. Considering the epidemiologic threat of prolonged CPE carriage, the potential value of our study may provide evidence to define high risk patients for implementing infection control strategies and antibiotic stewardship policies that can be applied especially in hospitals with CRE endemicity or resource-limited institutions. Moreover, our results may provide support for identifying the priority group of patients to perform FMT as a CPE decolonization strategy in future research.

## Figures and Tables

**Figure 1 jcm-10-00310-f001:**
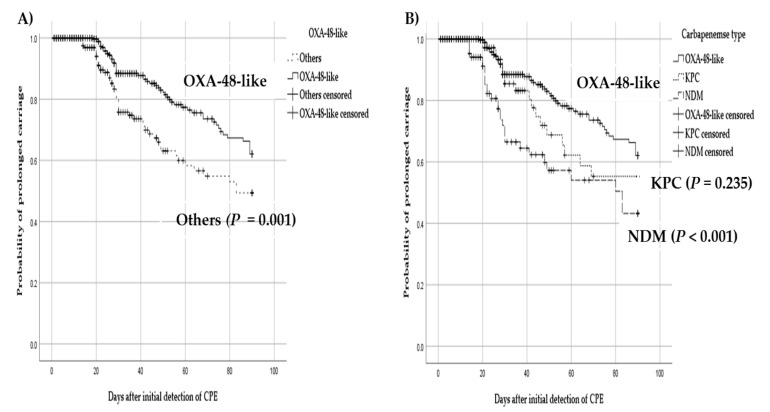
Kaplan–Meier estimates for the cumulative probability of prolonged carriage according to the types of carbapenemase. (**A**) OXA-48-like versus other types of carbapenemase gene (log-rank test); (**B**) stratified by three different types of carbapenemase gene (log-rank test). Abbreviations: OXA-48-like, Oxacillinase (OXA)-48-like; NDM, New-Delhi-Metallo-beta-lactamase; KPC, *Klebsiella pneumoniae* carbapenemase; CPE, carbapenemase-producing *Enterobacteriaceae*.

**Figure 2 jcm-10-00310-f002:**
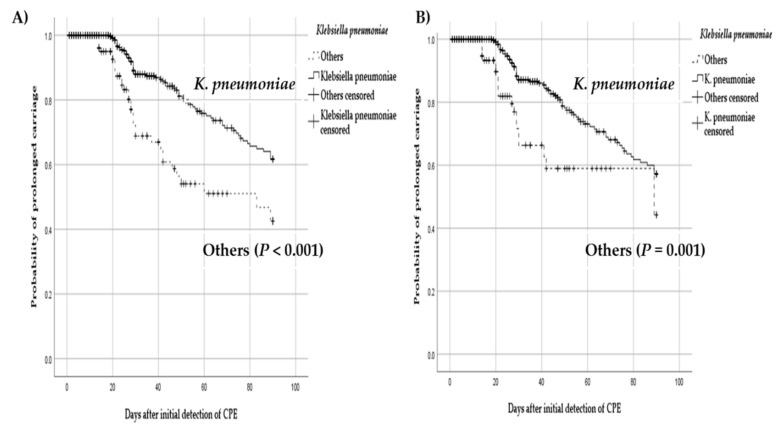
Kaplan–Meier estimates for the cumulative probability of prolonged carriage according to the types of organism. (**A**) *K. pneumoniae* versus other types of organism including patients with polymicrobial carriage (log-rank test); (**B**) *K. pneumoniae* versus other types of organism in patients with monomicrobial carriage (log-rank test).

**Figure 3 jcm-10-00310-f003:**
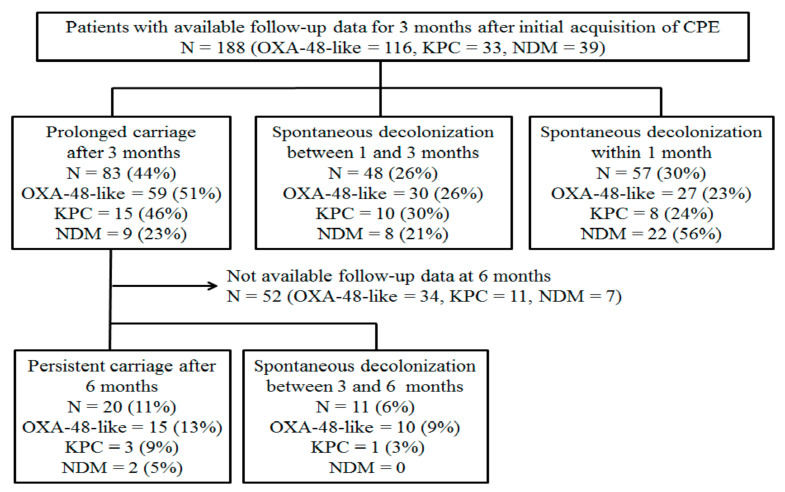
Flow chart of included patient with available follow-up culture data for 3 months after the initial detection of CPE to evaluate the clinical factors associated with prolonged carriage.

**Table 1 jcm-10-00310-t001:** Clinical characteristics of patients with carriage of carbapenemase-producing CRE who experienced spontaneous decolonization within 3 months or prolonged carriage after 3 months.

Variables	Decolonization within 3 Months ^a^ (*n* = 105)	Prolonged Carriage after 3 Months (*n* = 83)	*p*-Value
Age in years, median (IQR)	69 (53–78)	68 (54–79)	0.61
Male	60 (57)	43 (52)	0.47
Type of carbapenemase			
OXA-48-like	57 (54)	59 (71)	0.019
NDM	30 (29)	9 (11)	0.003
KPC	18 (17)	15 (18)	0.87
Type of organism			
Polymicrobial carriage ^b^	10 (10)	20 (24)	0.007
*K. pneumoniae*	83 (79)	81 (98)	<0.001
*E. coli*	1 (1)	13 (16)	<0.001
*Enterobacter* spp.	21 (20)	5 (6)	0.006
*C. freundii*	10 (10)	7 (8)	0.80
Underlying comorbidity			
Charlson’s score, median (IQR)	5 (3–7)	5 (3–7)	0.69
Immunosuppressive therapy ^c^	10 (10)	11 (13)	0.42
Surgery in the last 3 months	59 (56)	39 (47)	0.21
Catheter present ^d^	96 (91)	79 (95)	0.39
ICU admission ^e^	66 (63)	63 (76)	0.056
VRE colonization	25 (24)	35 (42)	0.007
*Clostridioides difficile* infection ^f^	12 (11)	24 (29)	0.002
Duration of index hospitalization, median days (IQR) ^g^	41 (24–60)	85 (56–119)	<0.001
Re-admission	34 (32)	20 (24)	0.21
Clinical positive culture	16 (15)	66 (81)	<0.001
Antibiotics use ^h^			
antipseudomonal penicillin	52 (50)	43 (52)	0.76
cephalosporin	54 (51)	48 (58)	0.38
carbapenem	39 (37)	63 (76)	<0.001
fluoroquinolones	37 (35)	44 (53)	0.015
amikacin	6 (6)	10 (12)	0.12
colistin	20 (19)	26 (31)	0.052
tigecycline	4 (4)	15 (18)	0.003
glycopeptide	52 (50)	59 (71)	0.003
In-hospital mortality	6 (6)	10 (12)	0.12

ICU: intensive care unit; IQR: interquartile range; VRE: vancomycin-resistant Enterococci. Data are presented as No. (%) unless indicated otherwise. ^a^ Three consecutive negative weekly surveillance cultures that were not followed by any positive cultures. ^b^ More than one isolated species of CPE had the same carbapenemase gene in the same patient. ^c^ Use of corticosteroid (≥20 mg of prednisolone or equivalent over 3 weeks, or ≥10 mg/day for at least 3 months), chemotherapy or radiotherapy, or other recognized T-cell immunosuppressants such as tumor necrosis factor-a blockers and calcineurin inhibitors during the 3 months of follow-up period. ^d^ Presence of any invasive catheter (central venous catheter, urinary catheter, enteral feeding tube, and tracheostomy) during the 3 months of follow-up period. ^e^ Resident in ICU more than 48 h during the 3 months of follow-up period. ^f^ Presence of symptoms and either a positive stool test result for *Clostridioides difficile* toxins, or colonoscopic findings demonstrating pseudomembranous colitis, during the 3 months of follow-up period. ^g^ Hospital stay after CPE identification. ^h^ Administration for more than 48 h after CPE identification, during the 3 months of follow-up period.

**Table 2 jcm-10-00310-t002:** Univariable and multivariable analyses of risk factors for prolonged carriage after 3 months among patients with carbapenemase-producing CRE colonization.

Variables ^a^	Univariable Analysis	Multivariable Analysis	
	OR (95% CI)	aOR (95% CI)	*p* Value
OXA-48-like	2.07 (1.12–3.81)	1.14 (0.18–7.18)	0.69
NDM	0.30 (0.14–0.68)		
Polymicrobial carriage	3.02 (1.32–6.87)		
*K. pneumoniae*	10.74 (2.45–47.14)	6.58 (1.05–41.27)	0.044
*E. coli*	19.31 (2.47–150.99)	13.48 (1.18–153.48)	0.036
*Enterobacter* spp.	0.26 (0.09–0.71)		
ICU admission	1.86 (0.98–3.53)		
VRE colonization	2.33 (1.25–4.36)		
*Clostridioides difficile* infection	3.15 (1.47–6.78)	3.98 (1.29–12.26)	0.016
Duration of index hospitalization, median days (IQR)	1.02 (1.01–1.03)	1.01 (1.004–1.03)	0.008
Clinical positive culture	21.60 (10.17–45.87)	11.14 (4.73–26.25)	<0.001
Carbapenem	5.33 (2.81–10.11)	2.32 (0.97–5.55)	0.06
Fluoroquinolones	2.07 (1.15–3.73)		
Colistin	1.94 (0.99–3.80)		
Tigecycline	5.57 (1.77–17.50)		
Glycopeptide	2.51 (1.36–4.61)		

aOR: adjusted odds ratio; CI: confidence interval; IQR: interquartile range; OR: odds ratio. ^a^ All variables with *p* value < 0.1 on univariable analysis were included in a multivariable logistic regression model.

**Table 3 jcm-10-00310-t003:** Univariable and multivariable analyses of risk factors for prolonged carriage after 3 months among the selected patients with OXA-48-like-producing CRE colonization (*n* = 116).

Variables ^a^	Univariable Analysis	Multivariable Analysis	
	OR (95% CI)	aOR (95% CI)	*p*-Value
*E. coli*	15.83 (1.99–125.55)		
Intensive care unit admission	2.02 (0.91–4.51)		
VRE colonization	2.32 (1.07–5.01)		
*Clostridioides difficile* infection	4.57 (1.56–13.34)	10.49 (2.01–54.88)	0.005
Duration of index hospitalization, median days (IQR)	1.02 (1.01–1.03)		
Clinical positive culture	18.25 (7.21–46.18)	13.93 (4.53–42.81)	<0.001
Carbapenem	10.27 (4.32–24.43)	8.96 (2.24–35.83)	0.002
Fluoroquinolones	2.35 (1.11–4.96)		
Colistin	2.69 (1.06–6.82)		
Tigecycline	4.13 (1.09–15.67)		
Glycopeptide	3.26 (1.49–7.13)		

^a^ All variables with *p* value < 0.1 on univariable analysis were included in a multivariable logistic regression model.

**Table 4 jcm-10-00310-t004:** Univariable and multivariable analyses of risk factors for prolonged carriage after 3 months among the selected patients with carbapenemase-producing *K. pneumoniae* colonization (*n* = 164).

Variables ^a^	Univariable Analysis	Multivariable Analysis	
	OR (95% CI)	aOR (95% CI)	*p*-Value
Intensive care unit admission	2.01 (1.03–3.93)		
VRE colonization	2.61 (1.32–5.18)		
*Clostridioides difficile* infection	4.57 (1.84–11.35)	4.38 (1.37–13.94)	0.013
Polymicrobial carriage	4.21 (1.59–11.12)		
Duration of index hospitalization, median days (IQR)	1.02 (1.01–1.03)	1.01 (1.00–1.03)	0.009
Clinical positive culture	17.01 (7.86–36.83)	10.40 (4.45–23.34)	<0.001
Carbapenem	6.08 (3.07–12.04)	2.95 (1.23–7.06)	0.015
Fluoroquinolones	2.00 (1.07–3.74)		
Colistin	2.40 (1.13–5.12)		
Tigecycline	6.06 (1.68–21.84)		
Glycopeptide	2.62 (1.36–5.02)		

^a^ All variables with *p* value < 0.1 on univariable analysis were included in a multivariable logistic regression model.

## Data Availability

The data presented in this study are available on request to the corresponding author on reasonable request.

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
