# Peer review of "Prolonged Carriage of Carbapenemase-Producing Enterobacteriaceae: Clinical Risk Factors and the Influence of Carbapenemase and Organism Types"

_jcm, 2021, doi:10.3390/jcm10020310_

Round 1
Reviewer 1 Report
This report underlines a possible prolonged carriage of OXA-48-like producers among the gut flora
- if any, this prolonged carriage shall focus only on Klebsiella pneumoniae
- ST type of each OXA-48 positive K. pneumoniae shall be at least determining as well as entire genome sequencing of each strain that belong to a same ST type. It cannot be exclude from the presented data that you are observing the spread of just a single clone
- Techniques for detecting carbapenemase producers are far to be optimum. Biochemical techniques shall be used instead
- molecular determination of the exact type of carbapenemase gene is mandatory. Maintenance of OXA-48 or OXA-181 may have not the same signification since their genetic support is not the same at all
- Did you identify carriage of the same carbapenemase gene in different species background in a sam patient ?
- in the discussion section, you have to give real hypotheses for what you are observing
Author Response
Comment 1: This report underlines a possible prolonged carriage of OXA-48-like producers among the gut flora. If any, this prolonged carriage shall focus only on Klebsiella pneumoniae. ST type of each OXA-48 positive K. pneumoniae shall be at least determining as well as entire genome sequencing of each strain that belong to a same ST type. It cannot be exclude from the presented data that you are observing the spread of just a single clone. Techniques for detecting carbapenemase producers are far to be optimum. Biochemical techniques shall be used instead. Molecular determination of the exact type of carbapenemase gene is mandatory. Maintenance of OXA-48 or OXA-181 may have not the same signification since their genetic support is not the same at all.
Response 1: Thank you for this valuable comment. As the reviewer’s comment, it could be more informative to analyze the different aspect of OXA-48 variants, according to the results of DNA sequencing. Although we did not investigate the ST type and the clonality of OXA-48-like CPE using methods such as MLST and DNA sequencing, we assumed that there have been clonal spread of OXA-48-like CPE in a region of our hospital. Molecular characteristics of CPE outbreaks in Korea includes OXA-232-producing K. pneumoniae in our hospital [SH Jeong et al. Diagn Microbiol Infect Dis 2015;82:70–2, SH Jeong et al. Ann Lab Med 2016;36:529–35]. However, our aim of the present study was to improve infection control approaches and strategies, while evaluating whether the prolonged carriage of CPE can be affected by the types of carbapenemase and organisms. Currently, there is paucity of knowledge about the carriage duration of OXA-48-like CPE, regardless of DNA sequencing results of OXA-48 variants. We believe that future research to evaluate the clonal spread of each OXA-48 variants using a technique in molecular biology may help regional coordination of CPE surveillance and control measures.
Following the reviewer’s comment, we have revised the manuscript as follow.
1) From) Discussion, P.11
Fourth, we believe that it would be worthwhile to include gut microbiome variables in the future in predicting prolonged carriage of CPE [35,40]. Despite these limitations, our study has some strength in that we included a relatively large number of patients to evaluate the clinical variables affecting prolonged carriage of distinct types of CPE.
To) Discussion, P.11
Fourth, molecular determination of the exact type of carbapenemase gene using MLST and entire genome sequencing of each strain was not performed in our study, although the genetic support may have different significations. We believe that future research to evaluate the clonal spread of CPE using a technique in molecular biology may help regional coordination of CPE surveillance and control measures. In addition, we believe that it would be worthwhile to include gut microbiome variables in the future in predicting prolonged carriage of CPE [35,40]. Despite these limitations, our study has some strength in that we included a relatively large number of patients to evaluate the clinical variables affecting prolonged carriage of distinct types of CPE.
Comment 2: Did you identify carriage of the same carbapenemase gene in different species background in a same patient? In the discussion section, you have to give real hypotheses for what you are observing.
Response 2: We agree with the reviewer’s concern. As the reviewer’s comment, we tried to find the presence of polymicrobial carriage that more than one isolated species of CPE had the same carbapenemase gene. In our study, only small number of patients colonized with CPE had the same carbapenemase gene in different species background in the same patient (6.1%, 43/702), including 39 K. pneumoniae with other organism at the same time. Polymicrobial carriage was not significantly associated with prolonged carriage in multivariable analysis, despite the statistical significance in univariable analysis. We believe that an additional study is fundamental to provide the potential impact of polymicrobial carriage on prolonged carriage. Therefore, based on the information of this study, we are planning the multicenter study with large scale in which we will provide the precise information about the impact of polymicrobial carriage on prolonged carriage.
Following the reviewer’s comment, we have revised the manuscript as follow.
1) To) added in Results, P.4
There were 659 patients with monobacterial colonization and 43 patients with polymicrobial carriage (more than one isolated species of CPE had the same carbapenemase gene in the same patient).
2) From) Discussion, P.10
Meanwhile, our results of the multivariable analysis of risk factors for prolonged carriage revealed that E. coli was also significantly associated with prolonged carriage. However, the clinical implication is uncertain, because of the small number of patients colonized with E. coli.
To) Discussion, P.10
Meanwhile, our results of the multivariable analysis of risk factors for prolonged carriage revealed that E. coli was also significantly associated with prolonged carriage. In addition, polymicrobial carriage was not significantly associated with prolonged carriage in multivariable analysis, despite the statistical significance in univariable analysis. However, the clinical implication is uncertain, because of the small number of patients colonized with E. coli and different species producing the same carbapenemase gene at the same time. We believe that an additional multicenter study with large scale is fundamental to provide the precise information about potential impact of them.
Thank you for your kind and helpful comments.
Reviewer 2 Report
Interesting and useful paper. One clarification: Did all pateints (with clinical CPE infection have positive index surveillance culture. It is not clear if all pateins were colonized fron the beginning, or initial colonization was inferred in clinical infections
Author Response
Interesting and useful paper. One clarification: Did all patients with clinical CPE infection have positive index surveillance culture? It is not clear if all patients were colonized from the beginning, or initial colonization was inferred in clinical infections.
Response 1: Thank you for this valuable comment. In our study, there were 27 patients with clinical CPE infection, and 11 of 27 patients had initial detection of CPE in clinical specimens. All of rectal swab cultures performed on the day of detection of clinical positive cultures revealed positive for CPE, which indicated the initial colonization prior to clinical infections. The rest of 16 of 27 patients had initial detection of CPE in universal active surveillance rectal swab cultures in ICUs, and suspicious clinical CPE infections occurred during their hospitalizations. To provide more specific information about the type of index culture in patients with clinical CPE infection, we tried to describe them in detail.
Following the reviewer’s comment, we have revised the manuscript as follow.
1) To) added in Results, P.6
The identification of CPE was found in 106 patients of rectal swab culture only and 82 patients of both rectal swab culture and clinical culture (46 sputum culture, 41 urine culture, 14 surgical wound culture, 9 blood culture, 7 intra-abdominal fluid culture). There were 27 patients with clinical positive culture, including bacteremia, treated for suspicious CPE infections. We found that 11 of 27 patients had initial detection of CPE in clinical specimens. All of rectal swab cultures performed on the day of detection of clinical positive cultures revealed positive for CPE, which indicated the initial colonization prior to clinical infections.
Thank you for your kind and helpful comments.
Reviewer 3 Report
In their retrospective, single-center study, Yong Kyun Kim and colleagues aimed to assess predictors of prolonged CPE carriage. I agree this endpoint is less frequently assessed than cross-transmission and outcome of CPE infection. This likely because of missing follow-up data, especially in retrospective studies (this limitation is also acknowledged by the authors with regard to the present paper).
Major comments
- Respectfully, I am somewhat confused by the employed analyses. First, the authors assessed predictors of prolonged carriage as a time-to-event endpoint, by univariable analysis through Kaplan-Meier methods. Two major limitations are the univariable nature of the test and the fact that the authors did not consider competing events, which are not taken into account by this method. Second, they used a fixed-effect logistic regression model for assessing prolonged carriage as a binary endpoint. Thus, it remains unclear to me why they assessed predictors of prolonged carriage twice with different methods. Furthermore, OXA-48 seemed not to be retained in the final multivariable logistic regression model (probably the authors employed a stepwise procedure, although this is not stated in methods). Thus OXA-48 was not independently associated with prolonged carriage according to this analysis, which is the contrary of what stated in the conclusions. Because of these apparent contradictions on core aspects of the paper, in my opinion the analyses should be reviewed by a statistician before proceeding further with revision of scientific contents.
Author Response
In their retrospective, single-center study, Yong Kyun Kim and colleagues aimed to assess predictors of prolonged CPE carriage. I agree this endpoint is less frequently assessed than cross-transmission and outcome of CPE infection. This likely because of missing follow-up data, especially in retrospective studies (this limitation is also acknowledged by the authors with regard to the present paper).
Comment 1: Respectfully, I am somewhat confused by the employed analyses. First, the authors assessed predictors of prolonged carriage as a time-to-event endpoint, by univariable analysis through Kaplan-Meier methods. Two major limitations are the univariable nature of the test and the fact that the authors did not consider competing events, which are not taken into account by this method. Second, they used a fixed-effect logistic regression model for assessing prolonged carriage as a binary endpoint. Thus, it remains unclear to me why they assessed predictors of prolonged carriage twice with different methods.
Response 1: Thank you for this valuable comment and we totally agree with the reviewer’s concern. In our study, large number of patients (514/702, 73.2%) were lost to follow-up or died within 3 months after the initial detection of CPE, which made us apply a multivariable logistic model to assess the strongest independent risk factor for prolonged carriage of CPE in the selected group of patients with available follow-up culture data for 3 months after initial detection of CPE. We assume that probabilities of prolonged carriage of CPE estimated by use of the Kaplan-Meier method and the log-rank test may help to assess the impact of types of carbapenemase and organism on prolonged carriage. However, as the reviewer’s comment, multivariable Cox proportional hazard regression analysis after adjustment of clinical variables should be performed to determine the impact of types of carbapenemase and organism on prolonged carriage. In this context, this study was limited to a preliminary study to determine the impact of types of carbapenemase and organism on prolonged carriage of CPE. Therefore, based on the results of this study, we are now planning the prospective multicenter study in which we will provide more refined information on the precise influence of types of carbapenemase and organism on prolonged carriage of CPE.
Following the reviewer’s comment, we have revised the manuscript as follow.
1) To) added in Discussion, P.11
Second, the retrospective nature of our present analyses may have resulted in information bias or missing data that influenced the results. We need to address that some patients did not have equal opportunities for assessment of decolonization in terms of number of total cultures and frequency of cultures (weekly during hospitalization, but not after discharge from hospital), which could introduce ascertainment bias because of the deviation from a standardized assessment. In addition, a considerable proportion of patients were lost to follow-up, which might have confounded the results of the risk factor analysis. Further well-designed prospective study in large scale to assess the impact of types of carbapenemase and organism on prolonged carriage of CPE by use of the multivariable Cox proportional hazard regression model after adjustment of clinical variables is clearly warranted. However, we gathered extensive clinical information and the infection control protocols were consistent during the study period.
Comment 2: Furthermore, OXA-48 seemed not to be retained in the final multivariable logistic regression model (probably the authors employed a stepwise procedure, although this is not stated in methods). Thus OXA-48 was not independently associated with prolonged carriage according to this analysis, which is the contrary of what stated in the conclusions. Because of these apparent contradictions on core aspects of the paper, in my opinion the analyses should be reviewed by a statistician before proceeding further with revision of scientific contents.
Response 2: Thank you for this valuable comment and we agree with the reviewer’s concern. Although OXA-48-like CPE showed the significantly increased probability of prolonged carriage in the Kaplan-Meier method and the log-rank test, multivariable logistic analysis revealed that OXA-48-like CPE was not independently associated with prolonged carriage in our study. As the reviewer’s comment, we tried not to stretch the meaning of the results in our study. Instead, based on the results of this study, we tried to provide more scientifically accurate information about the influence of OXA-48-like CPE on prolonged carriage. However, we believe that the results of this present study might help physicians in this area perform the further study to evaluate the duration of carriage of OXA-48-like CPE and the clinical risk factors associated with prolonged carriage in the patients with OXA-48-like CPE.
Following the reviewer’s comment, we have revised the manuscript as follow.
1) From) Abstract, P.1
The analyses of time to spontaneous decolonization using the Kaplan-Meier method showed that OXA-48-like and K. pneumoniae were significantly associated with prolonged carriage (log rank, P = 0.001 and P < 0.001). In risk factor analyses of the 188 patients with available follow-up culture data for 3 months after initial detection, multivariable analysis revealed that CPE positive clinical specimen (adjusted odds ratio [aOR] 11.14; 95% confidence interval [CI], 4.73–26.25; P < 0.001), concurrent Clostridioides difficile infection (CDI) (aOR 3.98, 95% CI 1.29–12.26; P = 0.016) were predictive of prolonged carriage. Our results suggest that specific type of CPE, such as OXA-48-like-producing K. pneumoniae, may have higher probability of prolonged carriage. Furthermore, patients with CPE in clinical specimen or concurrent CDI can cause a vicious circle in prolonged carriage.
To) Abstract, P.1
The analyses of time to spontaneous decolonization using the Kaplan-Meier method showed that OXA-48-like and K. pneumoniae were significantly associated with prolonged carriage (log rank, P = 0.001 and P < 0.001). In multivariable logistic analysis to assess the risk factors for CPE prolonged carriage in the 188 patients with available follow-up culture data for 3 months, K. pneumoniae (adjusted odds ratio [aOR] 6.58; 95% confidence interval [CI], 1.05–41.27; P = 0.044), CPE positive clinical specimen (aOR 11.14; 95% CI, 4.73–26.25; P < 0.001), concurrent Clostridioides difficile infection (CDI) (aOR 3.98, 95% CI 1.29–12.26; P = 0.016) were predictive of prolonged carriage. Our results suggest that K. pneumoniae may have higher probability of prolonged carriage, while the effect of OXA-48-like CPE is inconclusive. Furthermore, patients with CP-K. pneumoniae in clinical specimen or concurrent CDI can cause a vicious circle in prolonged carriage.
2) To) added in Table 2, P.8
Table 2. Univariable and multivariable analyses of risk factors for prolonged carriage after 3 months among patients with carbapenemase-producing CRE colonization.
|
|
Univariable analysis |
Multivariable analysis |
|
|
Variablesa |
OR (95% CI) |
aOR (95% CI) |
P value |
|
OXA-48-like |
2.07 (1.12–3.81) |
1.14 (0.18–7.18) |
0.69 |
|
NDM |
0.30 (0.14–0.68) |
|
|
|
Polymicrobial carriage |
3.02 (1.32–6.87) |
|
|
|
K. pneumoniae |
10.74 (2.45–47.14) |
6.58 (1.05–41.27) |
0.044 |
|
E. coli |
19.31 (2.47–150.99) |
13.48 (1.18–153.48) |
0.036 |
|
Enterobacter spp. |
0.26 (0.09–0.71) |
|
|
|
ICU admission |
1.86 (0.98–3.53) |
|
|
|
…….. |
|
|
|
3) From) Discussion, P.10
In the present study, we evaluated whether the types of carbapenemase and organism can affect the duration of carriage, and the clinical factors associated with prolonged carriage of CPE. Interestingly, the duration of CPE colonization was affected by types of carbapenemase and organism. Our results revealed that OXA-48-like-producing CPE and K. pneumoniae are more likely to have prolonged carriage compared to NDM CPE and other species such as Enterobacter spp., respectively. In addition, the results indicate that clinical positive culture, concurrent CDI, and longer duration of hospitalization were predictive of prolonged carriage of CPE.
To) Discussion, P.10
In the present study, we evaluated whether the types of carbapenemase and organism can affect the duration of carriage, and the clinical factors associated with prolonged carriage of CPE. Interestingly, the duration of CPE colonization was affected by types of carbapenemase and organism. The probability of prolonged carriage estimated by the Kaplan-Meier method showed that OXA-48-like-producing CPE and K. pneumoniae were more likely to have prolonged carriage compared to NDM CPE and other species such as Enterobacter spp., respectively. In addition, multivariable logistic analysis to assess the strongest independent risk factor for prolonged carriage of CPE revealed that K. pneumoniae was significantly associated with prolonged carriage, although OXA-48-like CPE was not significantly associated with prolonged carriage. Furthermore, the results indicate that clinical positive culture, concurrent CDI, and longer duration of hospitalization were predictive of prolonged carriage of CPE.
4) From) Conclusion, P.12
Our results suggest that specific types of carbapenemase and organism, such as OXA-48-like-producing K. pneumoniae, may have higher probability of prolonged carriage. In addition, patients with CPE in clinical specimens who are using carbapenem or have concurrent CDI can cause a vicious circle in prolonged carriage of CPE. Considering the epidemiologic threat of prolonged CPE carriage, the potential value of our study may provide evidence to define high risk patients for implementing infection control strategies and antibiotic stewardship policies that can be applied especially in hospitals with CRE endemicity or resource-limited institutions.
To) Conclusion, P.12
Our results suggest that K. pneumoniae may have higher probability of prolonged carriage, while the effect of OXA-48-like CPE on prolonged carriage is inconclusive. In addition, patients with CPE in clinical specimens who are using carbapenem or have concurrent CDI can cause a vicious circle in prolonged carriage of CPE. Considering the epidemiologic threat of prolonged CPE carriage, the potential value of our study may provide evidence to define high risk patients for implementing infection control strategies and antibiotic stewardship policies that can be applied especially in hospitals with CRE endemicity or resource-limited institutions.
Thank you for your kind and helpful comment
Round 2
Reviewer 3 Report
Thank you for your kind responses to comments. The last remaining concern is that, since the authors state there was a high mortality during follow-up, death may stand as a competing risk, thus the Kaplan Meier method may not be ideal for the assessment of prolonged carriage
Author Response
Comment 1: Thank you for your kind responses to comments. The last remaining concern is that, since the authors state there was a high mortality during follow-up, death may stand as a competing risk, thus the Kaplan Meier method may not be ideal for the assessment of prolonged carriage.
Response 1: Thank you for this valuable comment and we agree with the reviewer’s concern. In our study, large number of patients (514/702, 73.2%) were lost to follow-up or died within 3 months after the initial detection of CPE. Of these, 194 patients (194/514, 37.7%) died during the follow-up period, which was handled as censored data. As the reviewer’s comment, the research design of the present study to use the Kaplan-Meier method for the assessment of prolonged carriage of CPE despite the high rate of mortality might not be optimal. However, the main risk factors for CPE colonization include prior exposure to healthcare and broad-spectrum antibiotics [van Loon K et al. Antimicrob Agents Chemother 2017;62:e01730–17], and CPE colonization with or without subsequent CPE infection is associated with mortality rates as high as 50% [Tamma PD et al. Clin Infect Dis 2017;64:257–64, Dickstein Y et al. J Hosp Infect 2016;94:54–9, McConville TH et al. PLoS One 2017;12:e0186195]. The mortality rate of our study patients was comparable with that of previous studies, and the high rate of mortality in patients with CPE colonization and infection may be uncontrollable.
Nevertheless, to our knowledge, no study to evaluate the time to spontaneous decolonization of CPE by the different types of carbapenemase and organism has been conducted. Therefore, based on the results of this study, we believe that our future research of prospective multicenter study to use multivariable Cox proportional hazard regression analysis after adjustment of clinical variables to determine the impact of types of carbapenemase and organism on prolonged carriage should be performed.
We tried to adequately describe the limitation of the present study, and suggest more appropriate research design of the following study. Following the reviewer’s comment, we have revised the manuscript as follow.
1) To) added in Discussion, P.11
Second, the retrospective nature of our present analyses may have resulted in information bias or missing data that influenced the results. We need to address that some patients did not have equal opportunities for assessment of decolonization in terms of number of total cultures and frequency of cultures (weekly during hospitalization, but not after discharge from hospital), which could introduce ascertainment bias because of the deviation from a standardized assessment. In addition, a considerable proportion of patients were lost to follow-up or died, which might have confounded the results of the risk factor analysis and made the research design (the Kaplan-Meier method) not ideal for the assessment of prolonged carriage. Further well-designed prospective study in large scale to assess the impact of types of carbapenemase and organism on prolonged carriage of CPE by use of the multivariable Cox proportional hazard regression model after adjustment of clinical variables is clearly warranted. However, we gathered extensive clinical information and the infection control protocols were consistent during the study period.
Thank you for your kind and helpful comments.
